# Distinct roles of forward and backward alpha-band waves in spatial visual attention

**Andrea Alamia[1,2]\*, Lucie Terral[1], Malo Renaud D'ambra[1], Rufin VanRullen[1,2]**

[1]Cerco, CNRS Université de Toulouse, Toulouse, France; [2]Artificial and Natural Intelligence Toulouse Institute, Toulouse, France

**Abstract** Previous research has associated alpha-band [8–12 Hz] oscillations with inhibitory functions: for instance, several studies showed that visual attention increases alpha-band power in the hemisphere ipsilateral to the attended location. However, other studies demonstrated that alpha oscillations positively correlate with visual perception, hinting at different processes underlying their dynamics. Here, using an approach based on traveling waves, we demonstrate that there are two functionally distinct alpha-band oscillations propagating in different directions. We analyzed EEG recordings from three datasets of human participants performing a covert visual attention task (one new dataset with $N = 16$, two previously published datasets with $N = 16$ and $N = 31$). Participants were instructed to detect a brief target by covertly attending to the screen's left or right side. Our analysis reveals two distinct processes: allocating attention to one hemifield increases top-down alpha-band waves propagating from frontal to occipital regions ipsilateral to the attended location, both with and without visual stimulation. These top-down oscillatory waves correlate positively with alpha-band power in frontal and occipital regions. Yet, different alpha-band waves propagate from occipital to frontal regions and contralateral to the attended location. Crucially, these forward waves were present only during visual stimulation, suggesting a separate mechanism related to visual processing. Together, these results reveal two distinct processes reflected by different propagation directions, demonstrating the importance of considering oscillations as traveling waves when characterizing their functional role.

**\*For correspondence:**
artipago@gmail.com

**Competing interest:** The authors declare that no competing interests exist.

## Editor's evaluation

Alamia and colleagues investigate the direction of traveling waves in the α frequency band during visual spatial attention. The authors' novel perspective adopted here is important to understand the functional relevance of α oscillations for spatial attention. The observed pattern of results is consistent with distinct roles for travelling α waves in spatially opposite directions and makes a solid case for considering this new perspective on α rhythms in human cognitive function.

## Introduction

Brain oscillations are related to several cognitive functions, as they orchestrate neuronal activity at distinct temporal and spatial scales (*Buzsáki and Draguhn, 2004*; *Buzsáki, 2009*). Alpha-band oscillations [8–12 Hz] are the most prevailing rhythms in electrophysiological (EEG) recordings, spreading through most cortical regions.

Several studies investigated their functional role in various cognitive processes (*Palva and Palva, 2007*; *Palva and Palva, 2011*), providing mixed results. On the one hand, some studies showed that alpha-band oscillations might filter sensory information, regulating excitation and inhibition of

sensory-specific brain regions (*Jensen and Mazaheri, 2010*; *Mathewson et al., 2011*; *Klimesch, 2012*; *Sadaghiani and Kleinschmidt, 2016*). Accordingly, researchers interpreted alpha oscillations as a top-down mechanism involved in inhibitory control and timing of cortical processing (*Klimesch et al., 2007*), as well as modulating cortical excitability (*Jensen and Mazaheri, 2010*; *Mathewson et al., 2011*). Experimental studies corroborated this hypothesis, demonstrating how the phase of alpha-band oscillation affects visual perception (*Busch et al., 2009*; *Fakche et al., 2022*; but see also *Ruzzoli et al., 2019*). Another highly replicated result regarding the inhibitory role of alpha oscillations consists in the hemispheric modulation in occipital regions associated with visual attention, having an increase of power ipsilateral to the attended hemifield, and a corresponding decrease contralaterally (*Worden et al., 2000*; *Sauseng et al., 2005*; *Kelly et al., 2006*; *Thut et al., 2006*; *Händel et al., 2011*). On the other hand, other experimental studies have related alpha-band oscillations in occipital and parietal regions to perceptual processing and visual memory (*Bonnefond and Jensen, 2012*; *VanRullen, 2016*; *Pang et al., 2020*; *Luo et al., 2021*). For example, reverse-correlation techniques reveal that the visual system reverberates sensory information in the alpha-band for as long as 1 s, in what has been dubbed 'perceptual echoes' (*VanRullen and Macdonald, 2012*). Importantly, these echoes are a clear signature of sensory processing as they reflect the input's precise time course, are modulated by attention and have been dissociated from inhibitory alpha power modulation (*VanRullen and Macdonald, 2012*; *VanRullen, 2016*; *Brüers and VanRullen, 2018*; *Schwenk et al., 2020*).

Altogether, these experimental evidences support distinct and contradictory conclusions about alpha-band oscillation's functional role(s), which remains an open debate. Here, we address this question from a different perspective that interprets alpha-band oscillations as traveling waves (*Muller et al., 2018*; *Alamia and VanRullen, 2019*), thus considering their spatial component, and their propagation direction. Considering the case of visual attention, we tested the hypothesis that two functionally distinct alpha-band oscillations propagate along the frontal–occipital line in opposite directions. This compelling hypothesis about the different functional roles of alpha-band traveling waves derives from our previous studies (*Alamia and VanRullen, 2019*; *Pang et al., 2020*), in which we showed how visual perception modulates alpha waves, that is forward waves during visual stimulation, backward waves when the stimulus was off. In addition, this hypothesis is in line with previous studies suggesting that distinct alpha-band oscillations are related to specific cognitive processes (*Gulbinaite et al., 2017*; *Deng et al., 2019*; *Schuhmann et al., 2019*; *Sokoliuk et al., 2019*; *Kasten et al., 2020*). In this study, we analyzed three datasets, two publicly available (*Foster et al., 2017*; *Feldmann-Wüstefeld and Vogel, 2019*, see below), and one collected specifically for this study. In all datasets, participants attended either to the left or the right hemifield, while keeping central fixation. Our results confirmed the hemispheric modulation of alpha-band oscillations in posterior regions (*Worden et al., 2000*; *Sauseng et al., 2005*; *Kelly et al., 2006*; *Thut et al., 2006*; *Händel et al., 2011*) and revealed two distinct alpha-band traveling waves propagating in opposite directions. First, visual attention increases top-down alpha-band waves propagating from frontal to occipital regions ipsilateral to the attended location, and such waves correlate positively with alpha power in frontal and occipital regions. Moreover, our analysis demonstrates that visual attention also modulates contralateral forward waves, that is, waves propagating from occipital to frontal areas. Importantly, the attentional modulation of forward waves is crucially dependent on sustained sensory processing, as this modulation disappears in the absence of visual stimulation. In contrast, alpha-band top-down waves are present and modulated by visual attention irrespective of the presence or absence of concurrent sensory stimulation. These results demonstrate two distinct alpha-band oscillatory waves propagating in opposite directions, seemingly underlying different cognitive processes. The well-known lateralization effect observed in alpha-band can be interpreted as top-down traveling waves, and it is most likely related to inhibitory processes, in line with previous studies (*Jensen and Mazaheri, 2010*; *Händel et al., 2011*). However, different alpha-band oscillations propagate in a forward direction and are directly related to sensory processing, reconciling previous evidence linking alpha-band oscillations with visual processing (*VanRullen and Macdonald, 2012*; *Lozano-Soldevilla and VanRullen, 2019*).

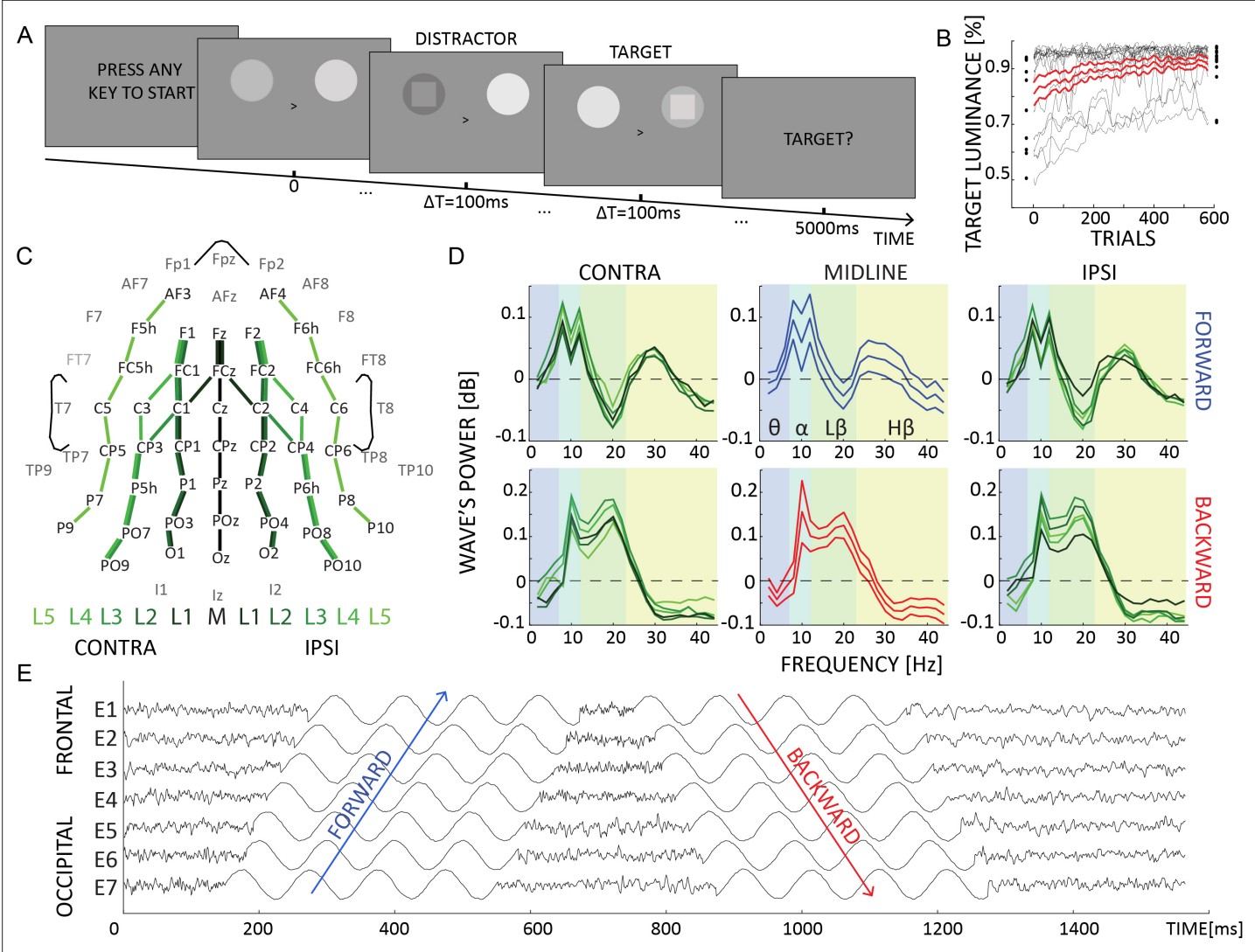

**Figure 1.** Experimental design and waves' spectral profile. (**A**) Each trial lasted 5 s, in which two flickering stimuli were presented to both hemifield. Participants were instructed to attend either the left or the right hemifield, as indicated by a central cue. In some trials, a target or a distractor appeared for 100 ms as a square either in the attended or unattended location. (**B**) The target and distractor luminance changed over trials due to the QUEST algorithm, which kept participants' performance around 80%. (**C**) We quantified traveling waves along 11 electrodes lines, running along the anterior–posterior axis. These lines were located in the contralateral or the ipsilateral hemisphere to the attended location. (**D**) The amount of waves in dB computed for forward (in blue) and backward (in red) waves in the midline (central subplots, thinner lines represent standard errors of the mean) and in the ipsi- and contralateral hemisphere (left and right panels, respectively). These waves were computed on trials without target or distractors. Positive (negative) values reflect more (less) waves than the chance level (as quantified by the surrogate distribution), whereas values around 0 indicate no difference between the real and the null distribution. (**E**) Simulated data providing a schematic representation of forward and backward waves in the time domain in a given line of electrodes (from more frontal E1 to more occipital E7). A positive and a negative phase shift characterized forward and backward waves, respectively.

## Results

### Traveling waves' spectral profile

The goal of the study was to investigate how visual attention modulates alpha-band traveling waves in the hemisphere contra- and ipsilateral to the attended location. To test this, we considered 11 lines of electrodes running from occipital to frontal regions (*Figure 1C*), 5 for each hemisphere and 1 midline. It is important to note that the spatial resolution of these lines is not critical for our analysis, as we do not expect significant differences within each hemisphere. However, before testing how visual attention modulates traveling waves, we explored the amount of waves propagating forward

(FW) and backward (BW) as a function of their temporal frequency (see Figure 5 and methods for a detailed description of the analysis). *Figure 1D* shows the spectral profile of FW and BW waves in the midline (along the Oz–Fz axis) and the contra- and ipsilateral lines: confirming previous experimental studies (*Alamia and VanRullen, 2019*; *Pang et al., 2020*), we found that alpha-band oscillatory waves propagate in both directions during visual stimulation, whereas theta (4–7 Hz) and high-beta/gamma (24–45 Hz) bands propagate mostly bottom-up from occipital to frontal regions, and low-beta (13–23 Hz) waves flow in the top-down direction. Interestingly, this pattern of results confirms previous studies using different methods, in which higher frequency bands (i.e., high-beta/gamma) have been associated with forward processing, whereas low-beta and alpha frequencies have been related to top-down processing (*Bastos et al., 2012*; *Bastos et al., 2015*; *van Kerkoerle et al., 2014*; *Michalareas et al., 2016*; but see also *Schneider et al., 2021b*).

## Attending to visual stimuli modulates traveling waves

In this analysis, we investigated how covert visual attention influences the traveling wave pattern. We focused on trials where neither a target nor a distractor was presented. First, we quantified the amount of traveling waves in the contra- and ipsilateral hemispheres to the attentional allocation. As shown in *Figure 2* (left column), we found a strong lateralization effect revealing an increase (respectively, decrease) of contralateral (ipsilateral) forward waves in the alpha-band, and the opposite pattern in waves propagating in the opposite direction. These results were confirmed by a Bayesian analysis of variance (ANOVA), considering as factors DIRECTION (FW or BW), LINES (distance from the midline), and LATERALITY (contra vs. ipsi). The results revealed strong evidence in favor of the interaction between DIRECTION and LATERALITY factors ($BF_{10}$ = 31.230, estimated error ~1%, $\eta^2$ = 0.08 as estimated from a classical ANOVA), whereas all other factors and their interactions revealed evidence in favor of the absence of an effect ($BFs_{10}$ < 0.3). We also found no significant effect in the other frequency bands (as shown in *Figure 1D*, namely theta, low, and high beta), hence we focused the following analyses on alpha-band oscillatory waves. These results demonstrate that the direction of alpha-band oscillatory traveling waves shows a laterality effect during a task involving covert selective attention.

## Backward waves correlate with alpha-band power

Previous studies investigating the role of alpha-band oscillations in visual attention reported a lateralization effect in the spectral power of alpha-band oscillations (*Worden et al., 2000*; *Sauseng et al., 2005*; *Kelly et al., 2006*; *Thut et al., 2006*; *Jensen and Mazaheri, 2010*; *Händel et al., 2011*). One may then wonder about the relationship between alpha power and traveling waves. To address this question, we investigated whether the oscillatory activity we observe propagating through the cortex relates to the spectral power in either occipital or frontal regions. We computed the averaged alpha-band power in frontal and occipital areas, contra- and ipsilaterally to the target presentation, considering the same electrodes used for quantifying the traveling waves (see *Figure 3A*). Interestingly, we found a significant positive correlation between alpha-band power in both occipital and frontal regions and backward waves, but not with forward waves (*Figure 3A* and *Table 1*). Next, we considered the lateralization effect in the alpha-band, as shown in *Figure 3B* (topographic plot in the right panel) and well replicated in previous studies (*Sauseng et al., 2005*; *Thut et al., 2006*; *Händel et al., 2011*). We wondered whether we could observe a correlation between such lateralization, defined as the difference between alpha-band power when attention is allocated to one side of the screen and to the other side, and the effect we reported in the traveling waves (*Figure 2*). As shown in *Figure 3B* (left panels), our results demonstrate a lack of correlation for both backward and forward waves in both frontal and occipital regions (all $|r|$ < 0.1, $BF_{10}$~ = 0.3).

To further investigate the relation between alpha-band traveling waves and alpha power, we performed the same analysis focusing on the correlation within each participant. In particular, we correlated trial-by-trial forward and backward waves with alpha-band power for each subject, obtaining correlation coefficients '*r*' and their respective p values. As in the previous analysis, we correlated forward and backward waves with frontal and occipital electrodes in both contro- and ipsilateral hemispheres. We applied the Fisher method (*Fisher, 1992*, see Methods for details) to combine all subjects' p values in every conditions. Overall, we found a significant effect of all combined p values (p < 0.0001), except in the lateralization condition (contra- minus ipsilateral hemisphere), similar to our previous analysis. Additionally, we tested for a

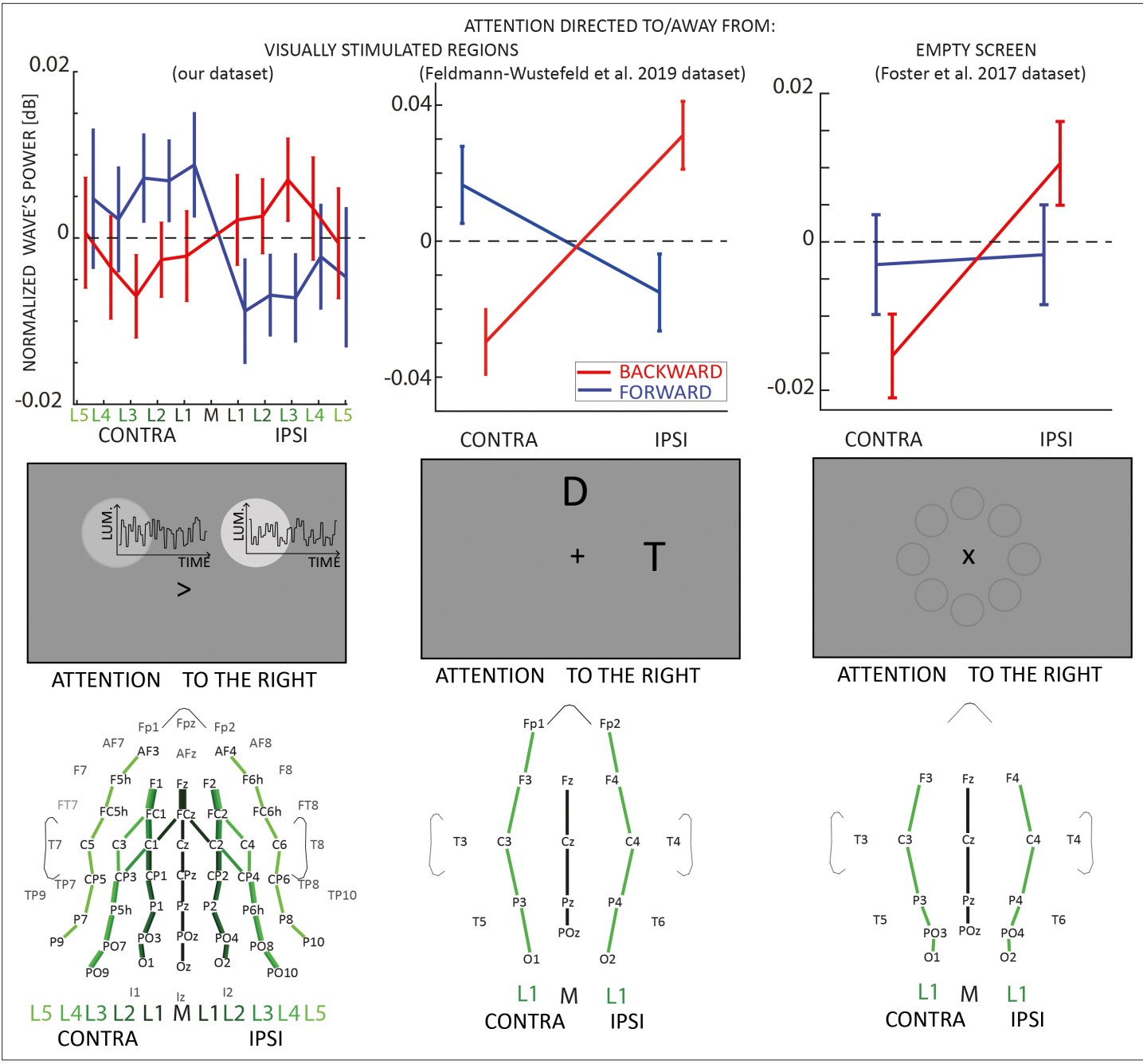

**Figure 2.** Traveling waves block analysis. Each column in the figure represents a different EEG dataset involving experiments with visual stimulation (left and middle columns) and without visual stimulation (right column). In the upper panels, the net amount of forward (blue) and backward (red) waves is represented along different lines of electrodes, normalized to the midline. The left and central panels reveal an increase (decrease) of forward (backward) waves contralateral to the attended location when participants attended to visual stimulation. The right column shows that when participants attended an empty screen (data from **Foster et al., 2017**), only backward waves were modulated by visual attention, and no effect was observed in the forward waves without visual stimulation. Error bars represent standard errors of the mean. The middle row shows schematic representations of the screen during the tasks: the central panel illustrates the task from **Feldmann-Wüstefeld and Vogel, 2019**, where D and T stand for Distractor and Target, respectively. In the task from **Foster et al., 2017**, the screen was empty, as the eight circles were not displayed during the task but here illustrate the stimulus positions (**Foster et al., 2017**). The lower panels represent the lines of electrodes in all datasets.

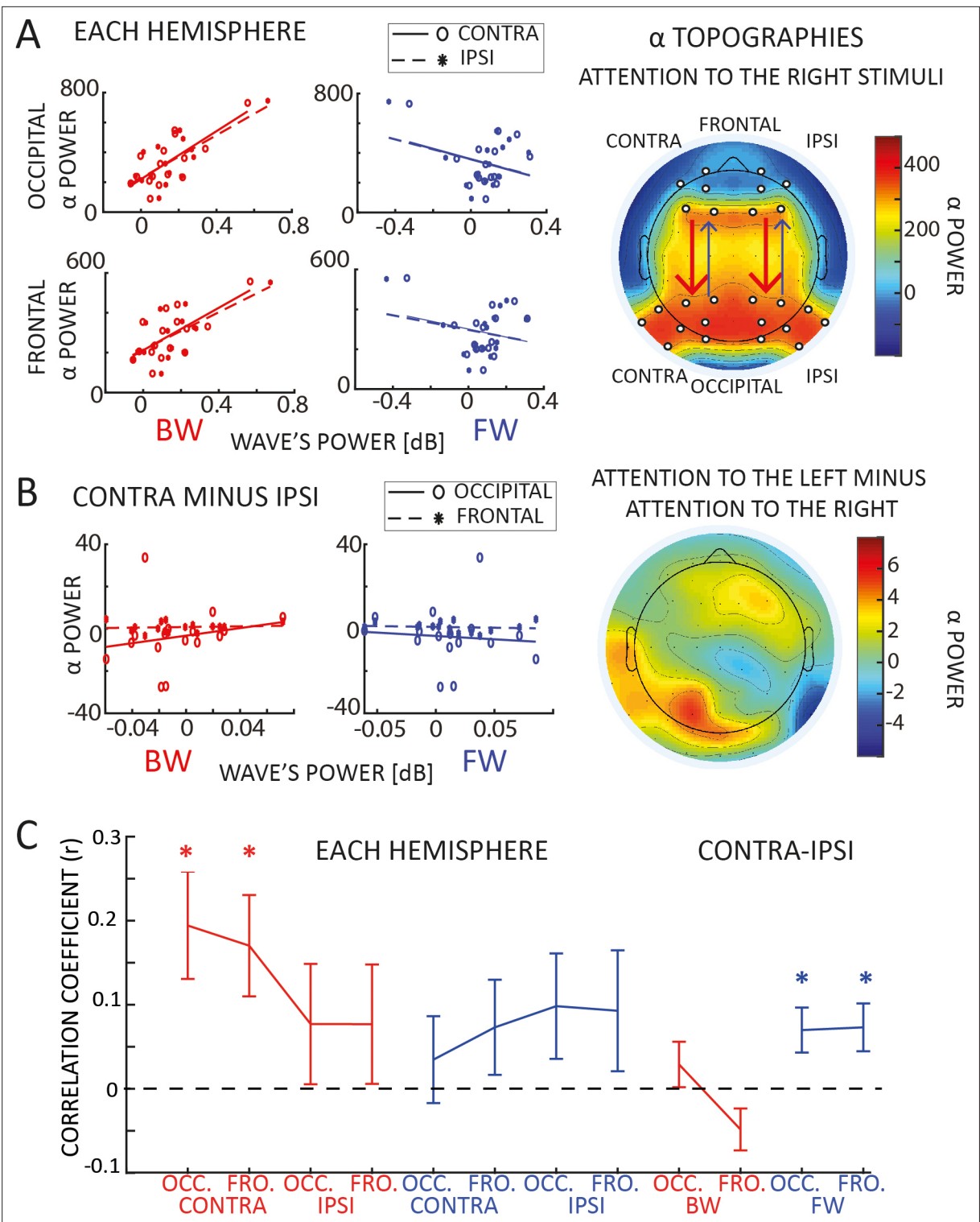

**Figure 3.** Correlation with alpha-band power. (**A**) Panel A reveals a correlation between backward waves and alpha power (static, standing power, i.e., measured via wavelets transform), in both frontal and occipital areas, in both hemispheres. We did not observe such correlation with forward waves. The plot to the right reveals the topographic distribution of alpha power when participants attended to the right hemifield (we included the 'left' condition by flipping the electrodes symmetrically to the midline). The white dots indicate the electrodes used for the correlation. (**B**) The plots to the left show the correlation between the laterality effect in the alpha power and in the waves (laterality measured as the mean difference between contra- and ipsilateral hemispheres for both alpha power and the waves – for the waves we computed the difference using lines of electrodes symmetrical to the midline). We did not observe any correlation in neither forward nor backward waves, with neither frontal nor occipital alpha power. The topography to

*Figure 3 continued on next page*

*Figure 3 continued*
the right reveals a lateralization effect in the alpha power (attention to the left minus attention to the right), confirming the presence of alpha power lateralization, in line with previous studies (**Sauseng et al., 2005**; **Thut et al., 2006**; **Händel et al., 2011**). (**C**) Panel C shows the trial-by-trial correlation coefficients averaged across participants for different conditions (as indicated in the x-axis). Confirming the results in panel A, we found a positive correlation across participants between backward waves and alpha power, specifically in the contralateral hemisphere. We also observed a positive global effect of the laterality condition across participants in the forward waves, even though the combined p values for the trial-by-trial correlation did not reach the significant threshold. Error bars represent standard errors of the mean.

consistent positive or negative distribution of the correlation coefficients. As shown in *Figure 3C*, the results support a significant correlation between backward waves and alpha power in the hemisphere contralateral to the attended location ($BF_{10}$ = 10.7 and $BF_{10}$ = 7.4 for occipital and frontal regions, respectively; all other $BF_{10}$ were between 1 and 2, providing inconclusive evidence). Interestingly, this analysis also revealed a small but consistent effect in the correlation between lateralization effects, as we reported a consistently positive correlation in the contra- minus ipsilateral difference between forward waves and alpha power ($BF_{10}$ ~ 5 for both frontal and occipital electrodes). However, it is important to notice that the combined p values obtained using the Fisher method did not reach the significance threshold in the lateralization condition, reducing the relevance of this specific result.

## Covert attention modulates forward waves only with visual stimuli

To confirm our previous results, we replicated the same traveling waves analysis on two publicly available EEG datasets in which participants performed similar attentional tasks (experiment 1 of *Foster et al., 2017* and experiment 1 of *Feldmann-Wüstefeld and Vogel, 2019*). In the first experiment from the Feldmann-Wüstefeld and Vogel dataset, participants were instructed to perform a visual working memory task in which, while keeping a central fixation, they had to memorize a set of items while ignoring a group of distracting stimuli. We focused our analysis on those trials in which the visual items to remember were placed either to the right or the left side of the screen, while the distractors were either in the upper or lower part of the screen (we pulled together the trials with either two or four distractors, as this factor was irrelevant for our analysis). The stimuli were shown for 200 ms, and we computed the amount of forward and backward waves in the 500 ms following stimulus onset. As shown in *Figure 2* (central column), the analysis confirmed our previous results, demonstrating a strong interaction between the factors DIRECTION and LATERALITY ($BF_{10}$ = 667, error ~2%; independently, the factors DIRECTION and LATERALITY had $BF_{10}$ = 0.2 and $BF_{10}$ = 0.4, respectively). These results confirmed that spatial attention modulates both forward and backward waves in the presence of visual stimulation. Next, we analyzed another publicly available dataset from *Foster et al., 2017*. In the first experiment of Foster's study, participants completed a spatial cueing task, requiring them to identify a digit among distractor letters. After a central cue was displayed for 250 ms, participants attended one of eight locations for 1000 ms before the onset of the target and distractors. As in our design, participants allocated attention to different locations to the left or right of the screen while keeping central fixation. However, unlike in our and in Feldmann-Wüstefeld's study, no stimulus was displayed while participants were attending one of the possible locations. Here, we assessed the amount of waves in the 1000 ms before the onset of the stimulus during attention allocation, when no visual stimuli were shown on the

**Table 1.** Correlation with alpha-band power.
The table reports the Pearson's correlation coefficient and the Bayes Factor ($BF_{10}$ supporting the alternative hypothesis, that is the presence of a correlation) between frontal and occipital electrodes and forward (FW) and backward (BW) waves, in both contra- and ipsilateral hemispheres. Values in bold reflect Bayes Factors providing strong evidence in favor of the alternative hypothesis. All correlations were computed on trials when neither a target nor a distractor was displayed.

| Pearson *r* ($BF_{10}$) | | FW | | BW | |
|---|---|---|---|---|---|
| | | **CONTRA** | **IPSI** | **CONTRA** | **IPSI** |
| OCC. | CONTRA | −0.297 (0.549) | −0.350 (0.697) | **0.720 (28.519)** | **0.698 (19.503)** |
| | IPSI | −0.305 (0.566) | −0.342 (0.669) | **0.786 (116.990)** | **0.746 (47.512)** |
| FRONT. | CONTRA | −0.222 (0.422) | −0.252 (0.465) | **0.772 (84.225)** | **0.712 (24.645)** |
| | IPSI | −0.327 (0.625) | −0.354 (0.710) | **0.747 (48.448)** | **0.705 (21.841)** |

screen. Remarkably, as shown in *Figure 2* (right panel), our analysis demonstrated an effect of the lateralization (LATERALITY: $BF_{10}$ = 3.571, error ~1%), revealing more waves contralateral to the attended location, but inconclusive results regarding the interaction between DIRECTION and LATERALITY ($BF_{10}$ = 2.056, error ~1%). However, using a classical ANOVA (i.e., without modeling the slope of the random terms), the interaction between DIRECTION and LATERALITY proved significant ($F(1,16)$ = 9.81, p = 0.003, $\eta^2$ = 0.13). In addition, when testing LATERALITY separately for forward and backward waves, we observed an effect in the backward waves ($BF_{10}$ = 3.497, error <0.01%) but not in the forward waves ($BF_{10}$ = 0.231, error <0.01%, supporting evidence in favor of the absence of an effect). In addition, as analyzed in our dataset, we tested the correlation between backward waves and alpha-band power in occipital (electrodes: PO3 and PO4) and frontal (electrodes: F3 and F4) regions. We found moderate evidence of a positive correlation between contra- and ipsilateral backward waves, and occipital (all Pearson's $r\sim$ = 0.4, all $BFs_{10}\sim$ = 3) but inconclusive evidence in the frontal areas (all Pearson's $r\sim$ = 0.3, all $BFs_{10}\sim$ = 2). These results supported those from our dataset, despite having a smaller amount of electrodes' lines, and potentially reduced statistical power (see *Figure 2*, lower panels). All in all, we could confirm our previous conclusion that covert visual attention modulates top-down oscillatory waves, showing this effect even in the absence of visual stimulation. In addition, we surmised that the lateralization effect we reported in the forward waves in our dataset (absent in the Foster dataset) is related to the steady visual stimulation during the attentional allocation, in line with our previous results demonstrating that oscillatory bottom-up waves reflect sensory processing (*Alamia and VanRullen, 2019*; *Pang et al., 2020*).

## Both detected targets and distractors elicit FW waves, but not missed targets

In our previous analysis, based on a subset of trials in which neither a target nor a distractor occurred, we demonstrated that sustained attention to one hemifield generates oscillatory alpha-band waves propagating forward in the contralateral hemisphere and backward in the ipsilateral one. We now assess whether the occurrence of a specific event, such as the onset of a target or a distractor stimulus, could induce the generation of transient oscillatory waves. For this reason, we replicated the same analysis on those trials including either a target or a distractor (on average, each participant performed 146.25 trials in each condition), to quantify the amount of waves locked to the onset of these events.

The upper panels of *Figure 4A* reveal the amount of forward and backward waves contralateral to the stimulus. Note that the targets and distractors appeared in the attended and unattended locations, respectively. A Bayesian ANOVA reveals no difference between targets and distractors (EVENT: $BF_{10}$ = 0.206, error ~1%), or their interaction (DIRECTION × EVENT: $BF_{10}$ = 0.423, error ~5%), as shown in the top-right panel of *Figure 4*. This result reveals that both target and distractor elicit forward waves propagating contralateral to the hemifield where they occur. Next, we investigated whether the waves in the hemisphere contralateral to the attended hemifield correlate with the participant's performance in detecting the target (a QUEST algorithm kept the accuracy throughout the experiment around 80%). Remarkably, we found an effect concerning the 'hit' and 'miss' target, as revealed by a significant interaction of the DIRECTION and EVENT factors (DIRECTION × EVENT: $BF_{10}$ = 4.085, error ~2%), as shown in the bottom-right panel of *Figure 4A*. Interestingly, *Figure 4B* reveals the amount of waves 400 ms before and after the onset of the stimulus, showing how a missed target is related to a decrease (increase) in forward (backward) waves contralateral (ipsilateral) to the attended location, possibly due to attentional fluctuations during each trial.

## Discussion

Previous studies demonstrated that selective attention modulates alpha-band oscillations in occipital and parietal regions (*Worden et al., 2000*; *Sauseng et al., 2005*; *Kelly et al., 2006*; *Thut et al., 2006*; *Händel et al., 2011*), supposedly indicating their involvement in top-down, inhibitory functions. Here, we took a novel perspective on these results by interpreting oscillations as traveling waves (*Muller et al., 2018*), thus considering their spatial component on top of the temporal one. Our results revealed two distinct alpha-band waves propagating in opposite directions: attention modulates oscillations traveling from occipital to frontal regions only in the presence of visual stimulation, thus relating forward waves to visual processing (*Lozano-Soldevilla and VanRullen, 2019*; *Pang et al., 2020*); whereas oscillations propagating in the opposite, top-down direction were modulated by attention irrespective of the presence or absence of concurrent visual stimulation; as in standard studies of alpha power lateralization (*Worden et al., 2000*; *Sauseng et al.,*

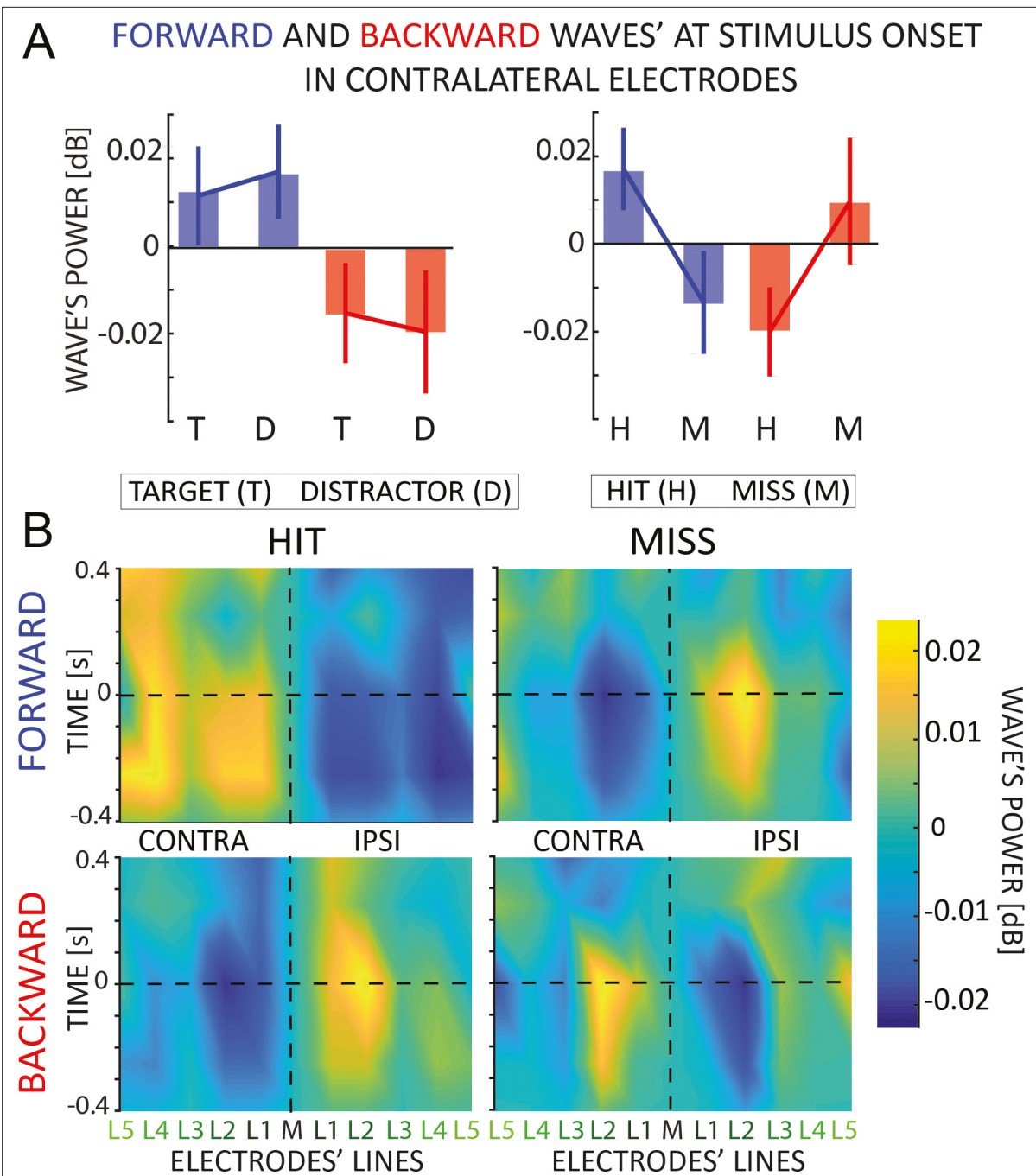

**Figure 4.** Event analysis. (**A**) The figure shows the amount of forward (in blue) and backward (in red) contralateral waves around the onset of the target/distractor (left) or hit and missed targets (right panel). Error bars are standard error of the mean. We found an interaction effect when we analyzed the hit versus missed target. (**B**) The 2D maps represent the amount of waves in the 11 lines of electrodes (x-axis) and around the onset time (y-axis) for forward and backward waves, and for hits and missed targets separately. The opposite pattern for hits versus misses, already visible before the target onset, suggests that missed targets are due to a failure of attentional allocation rather than sensory processing; and consequently, that proper attentional allocation is characterized by contralateral forward waves and ipsilateral backward waves.

*2005*; *Kelly et al., 2006*; *Thut et al., 2006*; *Händel et al., 2011*), this attentional modulation involved both an decrease of alpha waves contralateral to the attended location, and an ipsilateral increase.

In line with previous studies (*Gulbinaite et al., 2017*; *Deng et al., 2019*; *Schuhmann et al., 2019*; *Sokoliuk et al., 2019*; *Kasten et al., 2020*), our results support the thesis that distinct alpha-band oscillations are involved in separate cognitive processes. A recent study from *Sokoliuk et al., 2019* demonstrated two

different sources of alpha-band oscillations during a multisensory task: one, located in visual areas, reflects the 'spotlight of attention' and decreases linearly with increasing attention, whereas another one indicates attentional efforts and occurs in parietal regions. Gulbinaite et al. also demonstrated that parietal, but not occipital alpha-band oscillations are responsible for the oscillatory reverberation causing the 'triple-flash' illusion (*Gulbinaite et al., 2017*). Similarly, another study (*Kasten et al., 2020*) disentangled two primary sources of alpha oscillations, revealing a differential effect of tACS stimulation on endogenous but not exogenous attention. The authors interpreted their results as evidence supporting the hypothesis that alpha-band oscillations play a causal role in top-down but not bottom-up attention (*Schuhmann et al., 2019*; *Kasten et al., 2020*). Our results are consistent with these findings, including the spatial dimension in analyzing and interpreting alpha-band oscillations. Additionally, we also found a significant correlation between backward waves and occipital and frontal alpha-band power, consistently with Kasten's study (*Kasten et al., 2020*) and the inhibitory role of alpha-band oscillations. Our findings support the hypothesis that top-down processes, as reflected by backward waves, drive the well-documented hemispheric asymmetries reported in the literature (*Händel et al., 2011*; *Klimesch, 2012*; *Waldhauser et al., 2012*; *Peylo et al., 2021*). All in all, previous studies and our results pave the way for a more comprehensive understanding of the role of alpha oscillations in cognition.

One may wonder whether alpha-band oscillations during attention relate to target enhancement or distractor suppression (*Schneider et al., 2021a*). In the first case, the desynchronization of alpha activity would favor the sensory processing in the hemisphere contralateral to the target, whereas in the second case, alpha synchronization would suppress the processing of the distractor (*Kelly et al., 2006*; *Noonan et al., 2018*; *Peylo et al., 2021*). Our findings do not address this question directly but provide another element to the picture, suggesting the intriguing hypothesis that target enhancement is not reflected in the alpha power decrease but rather in an increase in the contralateral alpha-band waves processing the target and propagating forward. Our results thus support the hypothesis that alpha waves are involved in both distractor suppression (via ipsilateral top-down inhibitory feedback) and target enhancement (via contralateral bottom-up alpha-band waves). Future studies will precisely characterize the anatomical pathways of the distinct alpha-band oscillations, possibly involving cortical dynamics in the ventral and dorsal streams (*Capilla et al., 2014*) or corticothalamic connections (*Lopes da Silva et al., 1980*; *Halgren et al., 2019*).

Concerning the anatomical pathway of waves' propagation, our analysis based on EEG recordings prevents us from clearly determining whether the observed waves propagate through the cortex or whether more localized dipoles generate such widespread observations at the sensor level. A previous source-analysis study on different visual-task datasets (*Lozano-Soldevilla and VanRullen, 2019*) leaves both possibilities open. However, recent simulations on perceptual echo data (related to bottom-up, sensory waves, *Zhigalov and Jensen, 2022*) suggest that two dipoles in occipital and parietal regions could be responsible for the generation of the waves propagating in the occipital-to-frontal direction. Supposing this conclusion generalizes to raw EEG data and not only perceptual echoes (*VanRullen and Macdonald, 2012*), one could speculate that visual attention modulates dipoles selectively in each hemisphere. However, one may wonder whether similar dipoles are also responsible for generating top-down waves in frontal regions or whether other mechanisms are involved in generating alpha-band backward waves.

Our previous work proposed an alternative cause for the generation of cortical waves (*Alamia and VanRullen, 2019*). We demonstrated that a simple multilevel hierarchical model based on Predictive Coding (PC) principles and implementing biologically plausible constraints (temporal delays between brain areas and neural time constants) gives rise to oscillatory traveling waves propagating both forward and backward. This model is also consistent with the two-dipole hypothesis (*Zhigalov and Jensen, 2022*), considering the interaction between the parietal and occipital areas (i.e., a model of two hierarchical levels). However, dipoles in parietal regions are unlikely to explain the observed pattern of top-down waves, suggesting that more frontal areas may be involved in generating the feedback. This hypothesis is in line with the PC framework, in which top-down connections have an inhibitory function, suppressing the activity predicted by higher-level regions (*Huang and Rao, 2011*). Interestingly, Spratling proposed a simple reformulation of the terms in the PC equations that could describe it as a model of biased competition in visual attention, thus corroborating the interpretation of our finding within the PC framework (*Spratling, 2008*; *Spratling, 2012*).

In conclusion, our study demonstrated the existence of two functionally distinct alpha-band traveling waves propagating in opposite directions and modulated by visual attention. Top-down waves prevail in the hemisphere ipsilateral to the attended location and are related to inhibitory functions, whereas forward waves reflect ongoing visual processes in the contralateral hemisphere.

# Methods

## Participants

EEG data were recorded from 16 volunteers (aged 20–32 years old, four women, three left-handed). All subjects reported normal or corrected-to-normal vision, and they had no history of epileptic seizures or photosensitivity. All participants gave written informed consent before starting the experiment, following the Declaration of Helsinki. This study adheres to the guidelines for research at the 'Centre de Recherche Cerveau et Cognition', and the protocol was approved by the local ethical committee 'Comité de protection des Personnes Sud Méditerranée 1' (ethics approval number 2019-A02641-56). Furthermore, we included EEG recordings from two additional publicly available datasets investigating distinct scientific questions and using different analyses than our study. In the first one, 31 participants performed a visual working memory task involving spatial attention. The data were published in a previous study (*Feldmann-Wüstefeld and Vogel, 2019*, data available online at https://osf.io/a65xz/). In the second dataset, 16 participants performed a task involving covert spatial attention. These data were published in another study (*Foster et al., 2017*, data available online at https://osf.io/m64ue). The number of participants in our dataset was estimated based on a power analysis of previous studies investigating traveling waves in vision (*Luo et al., 2021*) and to match the number of participant in the third dataset (*Foster et al., 2017*). Our dataset is also available online at https://osf.io/pn784/.

## Experimental procedure

The following describes the experimental procedure to collect the data never published before. After setting the EEG device and placing the electrodes, participants performed the task in a dim and quiet room. The experiment was composed of 10 blocks of 60 trials each. During each trial (described in *Figure 1A*), two flickering luminance disks were displayed for 5 s, 9° to the left and right from the center of the screen. The flicker had a frequency of 160 Hz, and the intensities were pooled from a uniform distribution. We chose to apply flickering stimulation to keep participants engaged in the task and avoid attentional captures due to sudden target/distractor onset and offset (see below). Before each block, participants were instructed to allocate attention to either the right or the left stimulus while focusing on a central arrow located at the center of the screen. The arrow pointed to the attended location and served as a visual reminder throughout the block. In half of the trials, a target or a distractor flashed 100 ms inside the attended or non-attended stimulus (see *Figure 1A*). Their onset could occur any time after the first 500 ms of the trial. Both target and distractor were squares whose luminance was a percentage of the disk's luminance (i.e., when at 100%, targets/distractors were indiscernible from the disk, as they have the same luminance). A QUEST algorithm (*Watson and Pelli, 1983*) modulated such percentage to keep participants' performance around 80% throughout the experiment (see *Figure 1B*). In the other half of the trials, either the target followed the distractor's onset, or neither the target nor the distractor was presented (in sum, four possible trials occurred with the same frequency: either only a target, or only a distractor, or a target preceded by a distractor, or neither of them). Participants reported whether they saw a target only at each trial's end to prevent motor activity from confounding the EEG signals. All stimuli were generated in MATLAB using the Psychtoolbox (*Brainard, 1997*).

We included two additional datasets in this study. In both studies, participants performed a visual attention task while keeping their fixation in the center of the screen. Regarding the (*Feldmann-Wüstefeld and Vogel, 2019*) study, participants were asked to memorize the colors of two stimuli while ignoring a set of distractors stimuli. We analyzed uniquely those trials in which the visual stimuli were presented to the left or right side of the screen, while the distractors were placed above or below the fixation cross. After 500 ms of the fixation cross, two colored 'target' stimuli were presented for 200 ms. Participants were asked to memorize these stimuli, and a new 'probe' stimulus was shown after an additional second. Participants reported whether the probe matched the target stimuli or not. We analyzed the traveling waves in the 500 ms following the target stimulus onset.

Participants performed a spatial attention task in the second dataset from *Foster et al., 2017*. First, the fixation cross cued participants to covertly attend one of eight possible spatial positions uniformly distributed around the center of the screen. After 1 s, a digit was displayed either in the cued location or in any other one. The remaining locations were filled with letters. Participants were instructed to report the displayed digit. We analyzed the waves the second before the stimuli onset when participants were attending to the locations cued to the left or right side of the screen (we discarded trials in which participants attended locations above or below the fixation cross). For additional details about

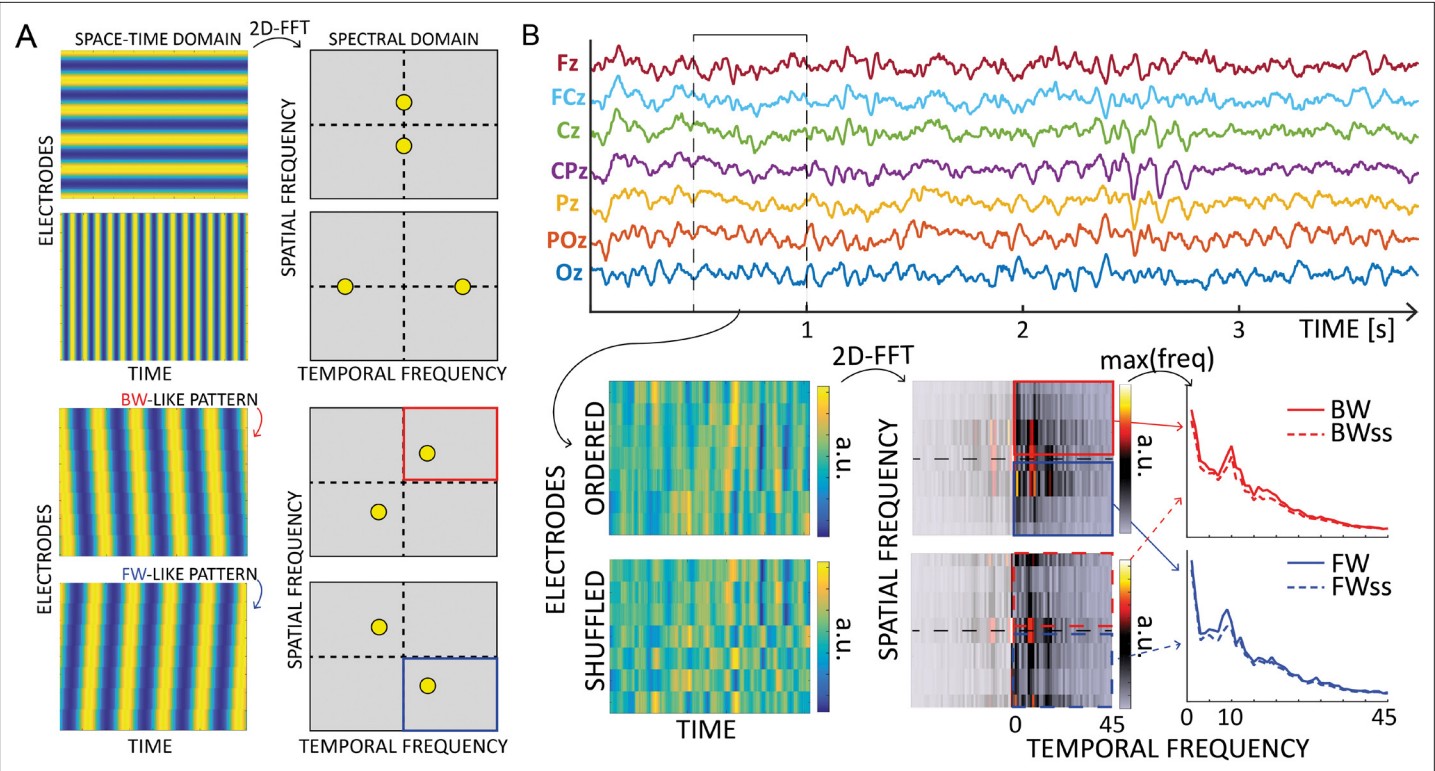

**Figure 5.** Waves analysis. (**A**) The 2D-Fast Fourier Trasform (2D-FFT) decomposes an image (e.g., a space-time representation of an EEG signal) into its spectral components. The upper part shows the decomposition of a 2D sinusoid propagating along the vertical or horizontal axis of the image. The corresponding peaks are found on the axis in the spectral domain, and their position depends on the frequency of the oscillations. The lower part of the figure shows how the spectra change when the oscillations propagate with a backward- or forward- like pattern. Importantly, the spectral peaks rotate in two of the four quadrants depending on the direction, providing a reliable measure of forward or backward waves in the image. (**B**) Schematic of the waves' quantification method. After defining time windows over each electrode line, we computed 2D Fourier transformation to quantify the amount of forward (in blue) and backward (in red) waves. From the upper and lower quadrants of the 2D-FFT spectra, we consider the maximum value over spatial frequencies, providing a 1D spectrum of forward and backward waves in the temporal frequency domain. The same procedure after shuffling the electrodes' order provides a surrogate measure, used as a baseline. Notably, such surrogate distribution captures the 1/f trend and the alpha-band peak, accounting for these factors in the final waves' quantification.

both experimental procedures, we refer the reader to *Foster et al., 2017* and *Feldmann-Wüstefeld and Vogel, 2019*.

## EEG recording and preprocessing

Throughout the experiment, we recorded EEG signals using a 64-channel active BioSemi EEG system (1024 Hz sampling rate), with three additional ocular electrodes. The preprocessing consisted of downsampling the data to 256 Hz, followed by a high-pass (>1 Hz) and a notch (47–53 Hz) filter. Data were then average-referenced and segmented from 500 ms before trial onset to its end. We performed the preprocessing in EEGlab (*Delorme and Makeig, 2004*). Importantly, we carefully discarded from all analyses all trials in which the EOG signals revealed eye movements.

## Traveling wave analysis

As in our previous studies (*Alamia et al., 2020*; *Pang et al., 2020*), we quantified traveling waves' propagation along 11 lines of seven electrodes, running from occipital to frontal regions. As shown in *Figure 1C*, we considered one midline (Oz, POz, Pz, CPz, Cz, FCz, and Fz) and five lines spreading through the left and right hemispheres, symmetrically to the midline. The electrodes' choice overlapped and covered a large portion of each hemisphere. For each set of seven electrodes, we created 2D maps by sliding a 500-ms time window over the EEG signals (having a 250-ms overlap) and computing 2D-FFT representations of each map (*Figure 5B*). Notably, the power in the lower and upper quadrants quantifies the amount of waves propagating forward (FW – from occipital to frontal electrodes) and backward (BW – from frontal to occipital),

respectively (see *Figure 5A*). Next, we performed the same procedure after shuffling the electrodes to obtain a baseline with the same spectral power but without information about the amount of FW/BW waves (note in *Figure 1D* that the surrogate distribution accounts for the typical $\frac{1}{f}$ power trend, as well as the alpha peak). Lastly, for each frequency in the range [2–45 Hz], we extracted the maximum values in the 2D-FFT spectra in both the real (FW and BW) and the shuffled data (FW$_{ss}$ and BW$_{ss}$), obtaining the waves' amount in decibel [dB] as:

$$FWwaves\left[\text{dB}\right] = 10 * \log_{10}\left(\frac{FW}{FW_{ss}}\right); BWwaves\left[dB\right] = 10 * \log_{10}\left(\frac{BW}{BW_{ss}}\right)$$

This value quantifies the total waves compared to the null distribution, thus being informative when contrasted against zero. Importantly, this waves analysis is limited to the surface level, and does not directly inform about the underlying sources. It is also necessary to keep in mind issues related to long-range connections and distortions due to scalp interference (*Nunez, 1974*; *Alexander et al., 2019*).

## Block analysis

This analysis quantified the amount of traveling waves across all trials when neither a target nor a distractor appeared. Participants paid attention to either one or the other side of the screen, thus defining a controlateral and an ipsilateral hemisphere (see *Figure 1C*). First, we averaged the forward and backward waves separately (in dB, see above) for each map computed along the 11 lines of electrodes within the 5-s trial (five lines for each hemisphere and the midline). Next, we averaged the results between trials, thus obtaining five contralateral and five ipsilateral spectra per subject for both FW and BW waves. Although our hypothesis focuses on alph-band oscillations, we also assessed the amount of waves in other frequency bands. Accordingly, from each spectrum, we computed the average per frequency band defined as ϑ (4–7 Hz), α (8–12 Hz), low β (13–24 Hz), and high β/γ (25–45 Hz). Besides following the frequency band definition found in the literature, such division reflects the waves' profile observed in the midline (*Figure 1D*). Then, we normalized each pair of symmetric lines (e.g., L1) by subtracting their mean value separately for each frequency band (i.e., $\frac{L_{i,contra}+L_{i,ipsi}}{2}$). This normalization allows to remove power differences across lines (e.g., L1, L2, etc.) and to compare the effects between hemispheres. Lastly, we tested an ANOVA considering as factors DIRECTION (either FW or BW), LINE (a value from 1 to 5 to define the distance from the midline), LATERALITY (contra- vs. ipsilateral), and all their interactions. We considered SUBJECTS as the random factor in the model. All models in this study relied on Bayesian statistics (see below for details). We performed the same analysis on the EEG recordings from the second dataset (16 participants performing a similar attentional task, see *Foster et al., 2017*, data available online at https://osf.io/m64ue). However, given the available data, we were able to consider only one electrodes' line per hemisphere, using the sensors O1-PO3-P3-C3-F3 and O2-PO4-P4-C4-F4 (see *Figure 2*, lower left panel).

## Waves and power correlation

This analysis assessed the correlation between FW and BW waves computed in the block analysis, with occipital and frontal alpha power. First, we estimated the mean alpha power in contra- and ipsilateral electrodes in both frontal and parieto-occipital regions, using the same electrodes involved in the waves' analysis (see *Figure 3B*). We computed power spectra using wavelet transform (1–45 Hz in log-space frequency steps with 1–20 cycles) for all trials when neither a target nor a distractor appeared. We then correlated the mean alpha power in both frontal and posterior regions with alpha-band forward and backward waves between subjects in both contra- and ipsilateral hemispheres. We reported Bayes Factor (BF) and Pearson's coefficients. Additionally, we computed trial-by-trial correlations between waves and alpha power for all participants. First, we tested the correlation coefficient against zero in all conditions. Then, we obtained a combined p value per condition using the log/lin regress Fisher method (*Fisher, 1992*), as shown in *Zoefel et al., 2019*. Specifically, we computed the *T* value of a chi-square distribution with 2*N degrees of freedom from the $p_i$ values of the *N* participants as:

$$T = -2 * \sum_{i=1}^{N} ln\left(p_i\right)$$

## Event analysis

In this analysis, we first investigated how the onset of a target or a distractor modulates the amount of both forward and backward waves, then whether a missed or correctly identified target elicits different patterns of waves. In both cases, we performed the same procedure as in the *block analysis*: we computed forward and backward waves separately for each line of electrodes obtaining five contralateral and five ipsilateral spectra per subject. First, we computed the waves 500 ms before and 500 ms after the target or distractor's onset, and we normalized each pair of symmetric lines as in the *block analysis* (see above). Then, we tested two separate ANOVAs considering in the first analysis the factor EVENT (either a target or a distractor occurred on the screen), and in the second the factor CORRECT (either a hit or a missed target) in the second analysis. We included DIRECTION (either FW or BW) in both models as a fixed factor and SUBJECTS as the random term.

## Statistical analyses

We computed BFs in all statistical analyses, measured as the ratio between the models testing the alternative against the null hypothesis. All BFs follow this indication throughout the paper and are denoted as $BF_{10}$. Conventionally, large BFs provide substantial (BF >~3) or strong (BF >~10) evidence in favor of the alternative hypothesis, whereas low BF (BF <~0.333) suggests a lack of effect (*Smith, 2001*; *Masson, 2011*). We performed all analyses in JASP (*Love et al., 2015*), considering default uniform prior distributions. The code to analyze the traveling waves is freely available at the following link: https://github.com/artipago/Travelling-waves-EEG-2.0 (*Alamia, 2023*).

## Acknowledgements

This work was funded by an ANR (OSCI-DEEP grant ANR-19-NEUC-0004) and ANITI (Artificial and Natural Intelligence Toulouse Institute) Research Chair (grant ANR-19-PI3A-0004) to R.V. A.A. has received funding from the European Research Council (ERC) under the European Union's Horizon 2020 research and innovation program (grant agreement no. 101075930).

## Additional information

### Funding

| Funder | Grant reference number | Author |
|---|---|---|
| European Research Council | 101075930 | Andrea Alamia |
| Agence Nationale de la Recherche | ANR-19-PI3A-0004 | Rufin VanRullen |

The funders had no role in study design, data collection, and interpretation, or the decision to submit the work for publication.

### Author contributions

Andrea Alamia, Conceptualization, Data curation, Formal analysis, Methodology, Writing - original draft; Lucie Terral, Malo Renaud D'ambra, Data curation, Methodology; Rufin VanRullen, Conceptualization, Supervision, Funding acquisition, Methodology, Project administration, Writing - review and editing

### Author ORCIDs

Andrea Alamia (iD) http://orcid.org/0000-0001-9826-2161
Rufin VanRullen (iD) http://orcid.org/0000-0002-3611-7716

### Ethics

This study adheres to the guidelines for research at the 'Centre de Recherche Cerveau et Cognition', and the protocol was approved by the local ethical committee 'Commité de protection des Personnes Sud Méditerranée 1' (ethics approval number 2019-A02641-56).

Decision letter and Author response
Decision letter https://doi.org/10.7554/eLife.85035.sa1
Author response https://doi.org/10.7554/eLife.85035.sa2

## Additional files

### Supplementary files
• MDAR checklist

### Data availability
Three datasets have been analyzed in this study and all of them are available from the Open Science Framework (as specified in the 'Methods/Participants' section). Code to analyze the data is available here: https://github.com/artipago/Travelling-waves-EEG-2.0 (copy archived at swh:1:rev:c7e4d66f63f-4c17c654bb258bc810874d53c51b3), as specified in the 'Methods/Statistical analysis' section.

The following dataset was generated:

| Author(s) | Year | Dataset title | Dataset URL | Database and Identifier |
|---|---|---|---|---|
| Alamia A | 2023 | Distinct roles of forward and backward alpha-band waves in spatial visual attention | https://osf.io/pn784/ | Open Science Framework, pn784 |

The following previously published dataset was used:

| Author(s) | Year | Dataset title | Dataset URL | Database and Identifier |
|---|---|---|---|---|
| Foster JJ, Sutterer DW, Serences JT, Vogel EK, Awh E | 2017 | Alpha-Band Oscillations Enable Spatially and Temporally Resolved Tracking of Covert Spatial Attention | https://doi.org/10.17605/OSF.IO/M64UE | Open Science Framework, 10.17605/OSF.IO/M64UE |

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
