## [Editor Report]

Alamia and colleagues investigate the direction of traveling waves in the α frequency band during visual spatial attention. The authors' novel perspective adopted here is important to understand the functional relevance of α oscillations for spatial attention. The observed pattern of results is consistent with distinct roles for travelling α waves in spatially opposite directions and makes a solid case for considering this new perspective on α rhythms in human cognitive function.

---

## [Decision Letter]

**Decision letter after peer review:**

Thank you for submitting your article "Distinct roles of forward and backward α-band waves in spatial visual attention" for consideration by *eLife*. Your article has been reviewed by 2 peer reviewers, and the evaluation has been overseen by a Reviewing Editor and Chris Baker as the Senior Editor. The following individual involved in the review of your submission has agreed to reveal their identity: Christian Keitel (Reviewer #1).

Essential revisions:

1) Although the results are in line with the conclusions drawn, some questions remain. The authors investigate the relationship between traveling α waves and the hemispheric lateralization of α power, which is a well-established neural signature of spatial attention. Surprisingly, the lateralization of α power shown in Figure 3B appears relatively weak in the present dataset (by visual inspection), which raises the question of whether the investigation of a relation between lateralized α power and α traveling waves is warranted in the first place.

(1b) Furthermore, the authors employ between-subject correlations (with N = 16) to test the relationship between α traveling waves and (lateralized) α power. However, as inter-individual differences in patterns of travelling waves are not the main focus here, within-subject analyses of the same relations would be able to test the authors' hypotheses much more directly.

2) For any naive reader the concept of travelling waves may be hard to grasp in the way data are currently presented – only based on the results of the 2D-FFT. Can forward and backward-travelling waves be illustrated in a representative example to make this more intuitive?

3) Findings from both datasets were difficult to discern and more effort should be made to highlight these. Also, a major conclusion "the directionality effect [effect of attention on forward waves] only occurs for visual stimulation" only rested on a qualitative comparison between studies. The authors have improved on this here, e.g., by toning down this conclusion. One thing that is still missing is a graphical representation of the data from Foster et al. (the second dataset analysed here) that would support the statistical results and allow the reader a visual comparison between the sets of findings.

4) Given that the Direction x Laterality interaction reported on page 9 was inconclusive (and likely not statistically significant) it is a bit difficult to interpret the apparent presence of an effect for forward waves only. Since the analysis pipeline appears to be very straightforward, it might be an option to support the claims by including one or two additional spatial attention datasets in the analysis.

In order to better understand the results, it would be helpful to show a temporally resolved plot of the power of forward and backward waves across the trial time course.

*Reviewer #1 (Recommendations for the authors):*

– Abstract: Mention "(N=16 each)" when introducing the studies.

– One idea to strengthen the "travelling" part of the wave would be to present a phase progression analysis of sorts that demonstrates a consistent phase difference in α oscillations along the "lines" of electrodes, potentially converted to physiologically plausible latencies that correspond to conduction delays. These could be of interest: Nunez et al. (2001) https://doi.org/10.1002/hbm.1030, Burkitt et al. 2000 https://doi.org/10.1016/S1388-2457(99)00194-7

– The Foster et al. dataset remains underspecified in the paper. I appreciate that this is available elsewhere but it would be great to provide some basic information somewhere in the paper. A separate section in the Methods that gives a few details on how the analysis differed from the 'main' dataset would be appreciated.

– For some of the analyses, data is broken down in increasingly smaller chunks based on target/distractor presence and behavioural outcome. However, I couldn't find information on how many trials were available per "condition" (on avg) for these. This seems important to convince the reader of an acceptable SNR for these analyses.

*Reviewer #2 (Recommendations for the authors):*

The Introduction is well written but the authors might want to structure the text into a few paragraphs (also applies to other sections of the manuscript).

At the end of the introduction, the authors state the hypothesis that two functionally distinct α-band oscillations propagate along the frontal-occipital line in opposite directions. It was a bit unclear where this hypothesis exactly comes from. The authors might want to explain the justification of this hypothesis in more detail or make a more balanced statement with respect to the question of whether this precise pattern was indeed hypothesized a priori.

I appreciate the reporting of Bayes Factors but since the present manuscript reports new findings (which will potentially be tested in new datasets in the future), it would be nice to report effect sizes as well.

In Figure 1D, it was a bit unclear to me what 0 (on the y-axis) refers to. The authors might want to report this in the caption.

The authors report between-subject correlation analyses (with N = 16) but as far as I understand, within-subject analyses (which are better powered) would test their claims more directly. Regarding α lateralization, it would be important to see whether the absence of a relation between lateralized α power and traveling waves' direction holds on a within-subject (single-trial) level. Between-subject correlations instead assume that there are robust and interpretable interindividual differences between subjects, which is at present unclear and possibly not what the authors want to demonstrate here.

The α lateralization in Figure 3B looks rather weak and it was not entirely clear to me what the unit (in colour) exactly is (probably a simple difference between attend-left and attend-right trials). Often, a lateralization index is used in the literature, which is more robust as it normalizes the lateralization for overall power. Also, I did not find a statistical test for this α lateralization pattern in the text. It would be important to demonstrate that a robust and statistically significant α lateralization is present in this dataset. Otherwise, it would be difficult to interpret the ensuing correlations of α lateralization with traveling waves.

Given that the Direction x Laterality interaction reported on page 9 was inconclusive (and likely not statistically significant) it is a bit difficult to interpret the apparent presence of an effect for forward waves only. Since the analysis pipeline appears to be very straightforward, it might be an option to support the claims by including one or two additional spatial attention datasets in the analysis.

In order to better understand the results, it would be helpful to show a temporally resolved plot of the power of forward and backward waves across the trial time course.

Page 11: "as in standard studies of α power lateralization (Worden et al., 2000; Sauseng et al., 2005; Kelly et al., 2006; Thut et al., 2006; Händel et al., 2011), this attentional modulation involved both an increase of α waves contralateral to the attended location, and an ipsilateral decrease". I guess it should be "decrease of α waves contralateral to the attended location, and an ipsilateral increase".

---

## [Author Response]

Essential revisions:1) Although the results are in line with the conclusions drawn, some questions remain. The authors investigate the relationship between traveling α waves and the hemispheric lateralization of α power, which is a well-established neural signature of spatial attention. Surprisingly, the lateralization of α power shown in Figure 3B appears relatively weak in the present dataset (by visual inspection), which raises the question of whether the investigation of a relation between lateralized α power and α traveling waves is warranted in the first place.

We agree with the reviewer that the effect seems reduced compared to other studies, although the topography of α-band lateralization in our data is in line with the literature. In order to quantify the effect, we performed an analysis similar to (Thut et al., 2006), defining a laterality index as:  indexa=α(ipsi ROI)−α(contra ROI)mean of α(contra+ipsi ROI)

We computed such index for occipital electrodes and their average (in red in Author response image 1). The results reveal that for most electrodes, including their average, the laterality index is significantly larger than 0, confirming the presence of α-band lateralization. However, we also note that the amplitude of the effect (~0.04) is reduced compared to the study by Thut and colleagues, which was between 0.05 and 0.10.

**Author response image 1. sa2fig1:** Laterality index for occipital electrodes, quantifying α-band lateralization during attention allocation. All electrodes go in the expected direction, revealing an increase of α-band power in the ipsilateral occipital hemisphere.

(1b) Furthermore, the authors employ between-subject correlations (with N = 16) to test the relationship between α traveling waves and (lateralized) α power. However, as inter-individual differences in patterns of travelling waves are not the main focus here, within-subject analyses of the same relations would be able to test the authors' hypotheses much more directly.

As suggested, we included the recommended within-subject analysis in the revised manuscript by computing a trial-by-trial correlation between α power and traveling waves for each participant. First, we obtained a correlation coefficient and a p-value for each subject. Then, we tested whether the correlation coefficients had an overall positive or negative distribution (i.e., according to our previous results, we expected a positive correlation between backward waves and α power). Additionally, we combined the p-values to test for overall significance (using the Fisher method, see Methods section below). Our results corroborate the between-subject correlation, supporting the conclusion that α-band power correlates mostly with backward waves (especially contro-lateral to the attended location). The other correlations (i.e., forward waves and α power) were statistically inconclusive. We included in the revised manuscript these new results, as shown in the following.

From the Results section:

"To further investigate the relation between α-band travelling waves and α power, we performed the same analysis focusing on the correlation within each participant. In particular, we correlated trial-by-trial forward and backward waves with α-band power for each subject, obtaining correlation coefficients' r' and their respective p-values. As in the previous analysis, we correlated forward and backward waves with frontal and occipital electrodes in both contro- and ipsilateral hemispheres. We applied the Fisher method (Fisher, 1992, see Methods for details) to combine all subjects' p-values in every conditions. Overall, we found a significant effect of all combined p-values (p<0.0001), except in the lateralization condition (contra- minus ipsilateral hemisphere), similar to our previous analysis. Additionally, we tested for a consistent positive or negative distribution of the correlation coefficients. As shown in figure 3C, the results support a significant correlation between backward waves and alphapower in the hemisphere contralateral to the attended location (BF_10_=10.7 and BF_10_=7.4 for occipital and frontal regions, respectively; all other BF_10_ were between 1 and 2, providing inconclusive evidence). Interestingly, this analysis also revealed a small but consistent effect in the correlation between lateralization effects, as we reported a consistently positive correlation in the contra- minus ipsilateral difference between forward waves and α power (BF_10_~5 for both frontal and occipital electrodes). However, it's important to notice that the combined p-values obtained using the Fisher method did not reach the significance threshold in the lateralization condition, reducing the relevance of this specific result. "

From the Methods section:

"Additionally, we computed trial-by-trial correlations between waves and α power for all participants. First, we tested the correlation coefficient against zero in all conditions. Then, we obtained a combined p-value per condition using the log/lin regress Fisher method (Fisher, 1992), as shown in (Zoefel et al., 2019). Specifically, we computed the T value of a chisquare distribution with 2*N degrees of freedom from the π values of the N participants as: T=−2∗∑i=1Nln(pi)

2) For any naive reader the concept of travelling waves may be hard to grasp in the way data are currently presented – only based on the results of the 2D-FFT. Can forward and backward-travelling waves be illustrated in a representative example to make this more intuitive?

We thank the reviewer for the suggestion. We included in figure 1 an additional panel E that represents a schematic example of forward and backward waves in the temporal domain (i.e., in the EEG data). We hope this example will provide a better understanding of the data and the traveling wave concept.

3) Findings from both datasets were difficult to discern and more effort should be made to highlight these. Also, a major conclusion "the directionality effect [effect of attention on forward waves] only occurs for visual stimulation" only rested on a qualitative comparison between studies. The authors have improved on this here, e.g., by toning down this conclusion. One thing that is still missing is a graphical representation of the data from Foster et al. (the second dataset analysed here) that would support the statistical results and allow the reader a visual comparison between the sets of findings.

We are glad that the reviewer recognizes the improvement in the presentation of the conclusions. According to their suggestions, we have modified figure 2, not only by including a third dataset (see point below) but also in a way that allows a direct comparison between the three datasets. Specifically, the results from the three datasets are now shown in three columns next to each other. The first row shows the FW and BW waves in contra and ipsilateral lines of electrodes: our dataset and the one from Feldmann-Wustefeld and colleagues (the first and the second column in the figure, both with visual stimulation) shows a clear interaction between direction and laterality, as confirmed by the statistical analysis. The dataset from Foster and colleagues (the third column, no visual stimulation) shows a laterality effect only in the backward waves but not in the forward ones, in line with the hypothesis that FW waves are modulated only in the presence of visual stimulation. The second row shows a schematic representation of the task, and the third row illustrates the electrodes' lines used in each dataset. We hope the reviewer will be satisfied with the current data presentation.

4) Given that the Direction x Laterality interaction reported on page 9 was inconclusive (and likely not statistically significant) it is a bit difficult to interpret the apparent presence of an effect for forward waves only. Since the analysis pipeline appears to be very straightforward, it might be an option to support the claims by including one or two additional spatial attention datasets in the analysis.

We have now included a third dataset in the paper. In this dataset, from (Feldmann-Wüstefeld and Vogel, 2019), participants performed a visual working memory task by attending either the left or the right side of the screen where a stimulus was displayed. We analyzed the amount of waves during stimulus presentation, and we found the same results as in our own dataset: very strong evidence in favor of an interaction between LATERALITY (contra- and ipsilateral) and DIRECTION (FW and BW). We now included the results in figure 2 (see point above) and in the Results section of the manuscript. Unfortunately, we couldn't find any other publicly available EEG dataset in which participants attend to either side of the screen without ongoing visual stimulation.

In addition, we re-analyzed our main findings (i.e. the interaction between LATERALITY and DIRECTION) in all three datasets using a classic ANOVA to report the effect size as η^2^ (see point above). Unlike the Bayesian ANOVA (which -in JASP- is based on linear mixed models), the classic one does not model the slope of the random effects. Yet, we observed that the LATERALITY x DIRECTION interaction in the Foster dataset proved very significant, with a large effect size (F(1,16)=9.81, p=0.003, η^2^=0.13). Supposedly, modeling the slope of the random effects in the Bayesian ANOVA lowered its statistical sensitivity. For the sake of completeness, we reported both results in the manuscript.

From the Results section:

“To confirm our previous results, we replicated the same traveling waves analysis on two publicly available EEG datasets in which participants performed similar attentional tasks (experiment 1 of Feldmann-Wüstefeld and Vogel, 2019 and experiment 1 of Foster et al., 2017). In the first experiment from the Feldmann-Wüstefeld and Vogel dataset, participants were instructed to perform a visual working memory task in which, while keeping a central fixation, they had to memorize a set of items while ignoring a group of distracting stimuli. We focused our analysis on those trials in which the visual items to remember were placed either to the right or the left side of the screen, while the distractors were either in the upper or lower part of the screen (we pulled together the trials with either 2 or 4 distractors, as this factor was irrelevant for the purposes of our analysis). The stimuli were shown for 200ms, and we computed the amount of forward and backward waves in the 500ms following stimulus onset. As shown in figure 2 (central column), the analysis confirmed our previous results, demonstrating a strong interaction between the factors DIRECTION and LATERALITY (BF_10_=667, error~2%; independently, the factors DIRECTION and LATERALITY had BF_10_=0.2 and BF_10_=0.4, respectively). These results confirmed that, in the presence of visual stimulation, spatial attention modulates both forward and backward waves. Next, we analyzed another publicly available dataset from Foster et al., 2017. […]”

“Remarkably, as shown in figure 2 (right panel), our analysis demonstrated an effect of the lateralization (LATERALITY: BF_10_=3.571, error~1%), revealing more waves contralateral to the attended location, but inconclusive results regarding the interaction between DIRECTION and LATERALITY (BF_10_=2.056, error~1%). However, using a classical ANOVA (i.e., without modeling the slope of the random terms), the interaction between DIRECTION and LATERALITY proved significant (F(1,16)=9.81, p=0.003, η^2^=0.13).”

From the Methods section:

“Furthermore, we included EEG recordings from two additional publicly available datasets investigating distinct scientific questions and using different analyses than our study. In the first one, 31 participants performed a visual working memory task involving spatial attention. The data were published in a previous study (Feldmann-Wüstefeld and Vogel, 2019, data available online at https://osf.io/a65xz/). In the second dataset, 16 participants performed a task involving covert spatial attention. These data were published in another study (Foster et al., 2017, data available online at https://osf.io/m64ue). The number of participants in our dataset was estimated based on a power analysis of previous studies investigating travelling waves in vision (Luo et al., 2021) and to match the number of participant in the third dataset (Foster et al., 2017).”

"We included two additional datasets in this study. In both studies, participants performed a visual attention task while keeping their fixation in the center of the screen. Regarding the Feldmann-Wüstefeld and Vogel, 2019 study, participants were asked to memorize the colors of two stimuli while ignoring a set of distractors stimuli. We analyzed uniquely those trials in which the visual stimuli were presented to the left or right side of the screen, while the distractors were placed above or below the fixation cross. After 500ms of the fixation cross, two colored 'target' stimuli were presented for 200ms. Participants were asked to memorize these stimuli, and a new 'probe’ stimulus was shown after an additional second. Participants reported whether the probe matched the target stimuli or not. We analyzed the traveling waves in the 500ms following the target stimulus onset.

Participants performed a spatial attention task in the second dataset from Foster et al. 2017. First, the fixation cross cued participants to covertly attend one of eight possible spatial positions uniformly distributed around the center of the screen. After one second, a digit was displayed either in the cued location or in any other one. The remaining locations were filled with letters. Participants were instructed to report the only displayed digit. We analyzed the waves the second before the stimuli onset when participants attended to the locations cued to the left or right side of the screen (we discarded trials in which participants attended locations above or below the fixation cross). For additional details about both experimental procedures, we refer the reader to Foster et al., 2017 and Feldmann-Wüstefeld and Vogel, 2019.”

In order to better understand the results, it would be helpful to show a temporally resolved plot of the power of forward and backward waves across the trial time course.

We thank the reviewer for the interesting suggestion. Author response image 2 shows the amount of forward and backward waves in the hemisphere contralateral to the attended location as a function of time. We considered only trials in which no target nor distractors were shown (note that figure 4B in the paper already shows the temporal dynamic around the onset of the target event). Overall the dynamics in Author response image 2 reveal a relatively constant dynamic, confirming the main results of figure 2 in the manuscript. Since we do not have a strong hypothesis or conclusion about the temporal dynamics shown in Author response image 2, we decided not to include it in the manuscript.

**Author response image 2. sa2fig2:** Amount of forward and backward waves in the hemisphere contralateral to the attended location as a function of time in trials with no targets nor distractors. Data are the average over the 16 subjects, ± standard errors of the mean.

Reviewer #1 (Recommendations for the authors):– Abstract: Mention "(N=16 each)" when introducing the studies.

We have included the datasets’ sizes in the revised abstract.

“We analyzed EEG recordings from three datasets of human participants performing a covert visual attention task (one new dataset with N=16, two previously published datasets with N=16 and N=31).”

– One idea to strengthen the "travelling" part of the wave would be to present a phase progression analysis of sorts that demonstrates a consistent phase difference in α oscillations along the "lines" of electrodes, potentially converted to physiologically plausible latencies that correspond to conduction delays. These could be of interest: Nunez et al. (2001) https://doi.org/10.1002/hbm.1030, Burkitt et al. 2000 https://doi.org/10.1016/S1388-2457(99)00194-7

We thank the reviewer for the suggestion. The updated figure 1 should now be helpful in clarifying the ‘travelling’ part of the waves, providing some examples. We have previously shown that the speed of waves measured with the 2D-FFT method are compatible with those measured at the scalp level with other techniques (see supplementary figure 3 of our previous work Alamia and VanRullen, (2019)). In this study, we have decided to apply the 2DFFT method because it proves, on average, more robust to noise (Alamia et al. 2020), as it considers all electrodes at once in a given time window (it would not be straightforward to have the same advantage using an approach based on the phases). The main limitation is that it sets a strong prior on the direction of propagation of the waves. This limitation is reduced in our study as we focus our analysis specifically on the frontal-occipital axis (in line with our hypothesis and previous studies).

– The Foster et al. dataset remains underspecified in the paper. I appreciate that this is available elsewhere but it would be great to provide some basic information somewhere in the paper. A separate section in the Methods that gives a few details on how the analysis differed from the 'main' dataset would be appreciated.

We thank the reviewer for the suggestion. We now included in the method section a paragraph explaining more in detail the experimental task. The analysis regarding the traveling waves is identical to the one performed on our dataset.

From the Methods section:

“We included two additional datasets in this study. In both studies, participants performed a visual attention task while keeping their fixation in the center of the screen. Regarding the Feldmann-Wüstefeld and Vogel, 2019 study, participants were asked to memorize the colors of two stimuli while ignoring a set of distractors stimuli. We analyzed uniquely those trials in which the visual stimuli were presented to the left or right side of the screen, while the distractors were placed above or below the fixation cross. After 500ms of the fixation cross, two colored ‘target’ stimuli were presented for 200ms. Participants were asked to memorize these stimuli, and a new ‘probe’ stimulus was shown after an additional second. Participants reported whether the probe matched the target stimuli or not. We analyzed the traveling waves in the 500ms following the target stimulus onset.

In the second dataset from Foster et al. 2017, participants performed a spatial attention task. First, the fixation cross cued participants to covertly attend one of eight possible spatial positions uniformly distributed around the center of the screen. After one second, a digit was displayed either in the cued location or in any other one. The remaining locations were filled with letters. Participants were instructed to report the displayed digit. We analyzed the waves the second before the stimuli onset when participants were attending to the locations cued to the left or right side of the screen (we discarded trials in which participants attended locations above or below the fixation cross). For additional details about both experimental procedures, we refer the reader to Foster et al., 2017 and Feldmann-Wüstefeld and Vogel, 2019.”

– For some of the analyses, data is broken down in increasingly smaller chunks based on target/distractor presence and behavioural outcome. However, I couldn't find information on how many trials were available per "condition" (on avg) for these. This seems important to convince the reader of an acceptable SNR for these analyses.

We thank the reviewer for pointing this out. According to the design, each participant performed 10 blocks, each composed of 60 trials. In each block, we present the target and the distractor alone in 15 trials each. Theoretically, across the experiment, each participant had 150 trials with the target only and 150 trials with the distractor only. In practice, due to eye movement rejection, each participant performed 146.25 trials for the trial condition (std = 5.9048, min=126, max=150) and 146.25 for the distractor condition (std = 3.9749, min=134, max=150). We included this information in the revised manuscript.

From the revised Results section:

“For this reason, we replicated the same analysis on those trials including either a target or a distractor (on average, each participant performed 146.25 trials in each condition), to quantify the amount of waves locked to the onset of these events.”

Reviewer #2 (Recommendations for the authors):The Introduction is well written but the authors might want to structure the text into a few paragraphs (also applies to other sections of the manuscript).

We thank the reviewer for the suggestion. We split the introduction into paragraphs accordingly (the Results section and the discussion are already separated in paragraphs).

At the end of the introduction, the authors state the hypothesis that two functionally distinct α-band oscillations propagate along the frontal-occipital line in opposite directions. It was a bit unclear where this hypothesis exactly comes from. The authors might want to explain the justification of this hypothesis in more detail or make a more balanced statement with respect to the question of whether this precise pattern was indeed hypothesized a priori.

We formulated this hypothesis starting from our previous studies, in which we showed that forward and backward waves were modulated differently by visual perception: specifically, we reasoned that bottom-up waves were most likely related to sensory processes, whereas backward waves relate to top-down functions, such as visual attention. In addition, we also got inspiration from previous studies showing distinct functional roles for different alphaband oscillations (as reported in the discussion). We have added these explanations in the revised version of the manuscript.

From the revised introduction:

“Considering the case of visual attention, we tested the hypothesis that two functionally distinct α-band oscillations propagate along the frontal-occipital line in opposite directions. This compelling hypothesis about the different functional roles of α-band traveling waves derives from our previous studies (Alamia and VanRullen, 2019; Pang et al., 2020), in which we showed how visual perception modulates α waves, i.e. forward waves during visual stimulation, backward waves when the stimulus was off. In addition, this hypothesis is in line with previous studies suggesting that distinct α-band oscillations are related to specific cognitive processes (Deng et al., 2019; Gulbinaite et al., 2017; Kasten et al., 2020; Schuhmann et al., 2019; Sokoliuk et al., 2019).”

I appreciate the reporting of Bayes Factors but since the present manuscript reports new findings (which will potentially be tested in new datasets in the future), it would be nice to report effect sizes as well.

We thank the reviewer for the suggestion. We included in the manuscript the effect sizes (η^2^) for the interactions between LATERALITY and DIRECTION (our main result) for each dataset, as shown in figure 2. Specifically, we observed a η^2^=0.08 for our dataset, η^2^=0.13 and η^2^=0.21 for the Foster et al. and for the Feldmann-Wustefeld et al. dataset, respectively.

In Figure 1D, it was a bit unclear to me what 0 (on the y-axis) refers to. The authors might want to report this in the caption.

We quantified the amount of waves as the log-ratio between the ‘real’ distribution and the surrogate one (i.e. obtained by shuffling the electrodes’ position). Accordingly, the results around 0 dB indicate no differences between the real and the null (surrogate) distribution, whereas positive (negative) results reflect more (less) waves than chance level (as quantified by the surrogate distribution). We modified the legend of the revised manuscript accordingly:

“(1D) […] Positive (negative) values reflect more (less) waves than the chance level (as quantified by the surrogate distribution), whereas values around 0 indicate no difference between the real and the null distribution.”

The authors report between-subject correlation analyses (with N = 16) but as far as I understand, within-subject analyses (which are better powered) would test their claims more directly. Regarding α lateralization, it would be important to see whether the absence of a relation between lateralized α power and traveling waves' direction holds on a within-subject (single-trial) level. Between-subject correlations instead assume that there are robust and interpretable interindividual differences between subjects, which is at present unclear and possibly not what the authors want to demonstrate here.

We included the suggested within-subject analysis in the revised manuscript. As suggested, we compute for each participant a trial-by-trial correlation between α power and traveling waves. For each participant, we obtained a correlation coefficient and a p-value. First, we tested whether the correlation coefficient had an overall positive or negative distribution (i.e. according to our previous results, we expected a positive correlation between backward waves and α power). Then, we combined the p-values to test for overall significance (using the Fisher method, see Methods section below). All in all our results corroborate the between-subject correlation, supporting the conclusion that α-band power correlate mostly with backward waves (especially contro-lateral to the attended location). The other correlation were statistically inconclusive. We included in the revised manuscript these new results, as shown in the following.

From the Results section:

"To further investigate the relation between α-band travelling waves and α power, we performed the same analysis focusing on the correlation within each participant. In particular, we correlated trial-by-trial forward and backward waves with α-band power for each subject, obtaining correlation coefficients' r' and their respective p-values. As in the previous analysis, we correlated forward and backward waves with frontal and occipital electrodes in both contro- and ipsilateral hemispheres. We applied the Fisher method (Fisher, 1992, see Methods for details) to combine all subjects' p-values in every conditions. Overall, we found a significant effect of all combined p-values (p<0.0001), except in the lateralization condition (contra- minus ipsilateral hemisphere), similar to our previous analysis. Additionally, we tested for a consistent positive or negative distribution of the correlation coefficients. As shown in figure 3C, the results support a significant correlation between backward waves and alphapower in the hemisphere contralateral to the attended location (BF_10_=10.7 and BF_10_=7.4 for occipital and frontal regions, respectively; all other BF_10_ were between 1 and 2, providing inconclusive evidence). Interestingly, this analysis also revealed a small but consistent effect in the correlation between lateralization effects, as we reported a consistently positive correlation in the contra- minus ipsilateral difference between forward waves and α power (BF_10_~5 for both frontal and occipital electrodes). However, it's important to notice that the combined p-values obtained using the Fisher method did not reach the significance threshold in the lateralization condition, reducing the relevance of this specific result. "

From the Methods section:

"Additionally, we computed trial-by-trial correlations between waves and α power for all participants. First, we tested the correlation coefficient against zero in all conditions. Then, we obtained a combined p-value per condition using the log/lin regress Fisher method (Fisher, 1992), as shown in (Zoefel et al., 2019). Specifically, we computed the T value of a chisquare distribution with 2*N degrees of freedom from the π values of the N participants as: T=−2∗∑i=1Nln(pi)

The α lateralization in Figure 3B looks rather weak and it was not entirely clear to me what the unit (in colour) exactly is (probably a simple difference between attend-left and attend-right trials). Often, a lateralization index is used in the literature, which is more robust as it normalizes the lateralization for overall power. Also, I did not find a statistical test for this α lateralization pattern in the text. It would be important to demonstrate that a robust and statistically significant α lateralization is present in this dataset. Otherwise, it would be difficult to interpret the ensuing correlations of α lateralization with traveling waves.

We agree with the reviewer that, despite the topography of α-band lateralization in our data is in line with the literature, the effect seems rather smaller than in other studies. In order to quantify the effect, we performed an analysis similar to (Thut et al., 2006), defining a laterality index as:  indexa=α(ipsi ROI)−α(contra ROI)mean of α(contra+ipsi ROI)

We computed such index for occipital electrodes, and on their average. Author response image 1 reveals that for most electrodes, including their average, the laterality index is significantly larger than 0, confirming the presence of an α-band lateralization effect. However, we also note that the size of the effect, around 0.04, is reduced as compared to the study by Thut and colleagues, which was between 0.05 and 0.10.

Given that the Direction x Laterality interaction reported on page 9 was inconclusive (and likely not statistically significant) it is a bit difficult to interpret the apparent presence of an effect for forward waves only. Since the analysis pipeline appears to be very straightforward, it might be an option to support the claims by including one or two additional spatial attention datasets in the analysis.

We have now included a third dataset in the paper. In this dataset, from (Feldmann-Wüstefeld and Vogel, 2019), participants performed a visual working memory task by attending either the left or the right side of the screen where a stimulus was displayed. We analyzed the amount of waves during stimulus presentation, and we found the same results as in our own dataset: very strong evidence in favor of an interaction between LATERALITY (contra- and ipsilateral) and DIRECTION (FW and BW). We now included the results in figure 2 (see point above) and in the Results section of the manuscript. Unfortunately, we couldn't find any other publicly available EEG dataset in which participants attend to either side of the screen without ongoing visual stimulation.

In addition, we re-analyzed our main findings (i.e. the interaction between LATERALITY and DIRECTION) in all three datasets using a classic ANOVA to report the effect size as η^2^ (see point above). Unlike the Bayesian ANOVA (which -in JASP- is based on linear mixed models), the classic one does not model the slope of the random effects. Yet, we observed that the LATERALITY x DIRECTION interaction in the Foster dataset proved very significant, with a large effect size (F(1,16)=9.81, p=0.003, η^2^=0.13). Supposedly, modeling the slope of the random effects in the Bayesian ANOVA lowered its statistical sensitivity. For the sake of completeness, we reported both results in the manuscript.

From the Results section:

“To confirm our previous results, we replicated the same traveling waves analysis on two publicly available EEG datasets in which participants performed similar attentional tasks (experiment 1 of Foster et al., 2017 and experiment 1 of Feldmann-Wüstefeld and Vogel, 2019). In the first experiment from the Feldmann-Wüstefeld and Vogel dataset, participants were instructed to perform a visual working memory task in which, while keeping a central fixation, they had to memorize a set of items while ignoring a group of distracting stimuli. We focused our analysis on those trials in which the visual items to remember were placed either to the right or the left side of the screen, while the distractors were either in the upper or lower part of the screen (we pulled together the trials with either 2 or 4 distractors, as this factor was irrelevant for the purposes of our analysis). The stimuli were shown for 200ms, and we computed the amount of forward and backward waves in the 500ms following stimulus onset. As shown in figure 2 (central column), the analysis confirmed our previous results, demonstrating a strong interaction between the factors DIRECTION and LATERALITY (BF_10_=667, error~2%; independently, the factors DIRECTION and LATERALITY had BF_10_=0.2 and BF_10_=0.4, respectively). These results confirmed that, in the presence of visual stimulation, spatial attention modulates both forward and backward waves. Next, we analyzed another publicly available dataset from Foster et al., 2017. […]”

“Remarkably, as shown in figure 2 (right panel), our analysis demonstrated an effect of the lateralization (LATERALITY: BF_10_=3.571, error~1%), revealing more waves contralateral to the attended location, but inconclusive results regarding the interaction between DIRECTION and LATERALITY (BF_10_=2.056, error~1%). However, using a classical ANOVA (i.e., without modeling the slope of the random terms), the interaction between DIRECTION and LATERALITY proved significant (F(1,16)=9.81, p=0.003, η^2^=0.13).”

From the Methods section:

"We included two additional datasets in this study. In both studies, participants performed a visual attention task while keeping their fixation in the center of the screen. Regarding the Feldmann-Wüstefeld and Vogel, 2019 study, participants were asked to memorize the colors of two stimuli while ignoring a set of distractors stimuli. We analyzed uniquely those trials in which the visual stimuli were presented to the left or right side of the screen, while the distractors were placed above or below the fixation cross. After 500ms of the fixation cross, two colored 'target' stimuli were presented for 200ms. Participants were asked to memorize these stimuli, and a new 'probe’ stimulus was shown after an additional second. Participants reported whether the probe matched the target stimuli or not. We analyzed the traveling waves in the 500ms following the target stimulus onset.

Participants performed a spatial attention task in the second dataset from Foster et al. 2017. First, the fixation cross cued participants to covertly attend one of eight possible spatial positions uniformly distributed around the center of the screen. After one second, a digit was displayed either in the cued location or in any other one. The remaining locations were filled with letters. Participants were instructed to report the only displayed digit. We analyzed the waves the second before the stimuli onset when participants attended to the locations cued to the left or right side of the screen (we discarded trials in which participants attended locations above or below the fixation cross). For additional details about both experimental procedures, we refer the reader to Foster et al., 2017 and Feldmann-Wüstefeld and Vogel, 2019.”

In order to better understand the results, it would be helpful to show a temporally resolved plot of the power of forward and backward waves across the trial time course.

We thank the reviewer for the interesting suggestion. Author response image 2 shows the amount of forward and backward waves in the hemisphere contralateral to the attended location as a function of time. We considered only trial in which no target nor distractors were shown (figure 4B in the paper already shows the temporal dynamic around the onset of the target event). Overall the dynamics in Author response image 2 reveal a relatively constant dynamic, confirming the main results of figure 2 in the manuscript. Since we do not have a strong hypothesis or conclusion about the temporal dynamics shown in Author response image 2, we decided not to include it in the manuscript.

Page 11: "as in standard studies of α power lateralization (Worden et al., 2000; Sauseng et al., 2005; Kelly et al., 2006; Thut et al., 2006; Händel et al., 2011), this attentional modulation involved both an increase of α waves contralateral to the attended location, and an ipsilateral decrease". I guess it should be "decrease of α waves contralateral to the attended location, and an ipsilateral increase".

Thanks for the correction, we have edited the sentence in the revised manuscript accordingly.

References

Alamia, A., and VanRullen, R. (2019). Α oscillations and traveling waves: Signatures of predictive coding? *PLOS Biology*, *17*(10), e3000487. https://doi.org/10.1371/journal.pbio.3000487

Deng, Y., Reinhart, R. M. G., Choi, I., and Shinn-Cunningham, B. (2019). Causal links between parietal α activity and spatial auditory attention. *ELife*. https://doi.org/10.7554/*eLife*.51184

Feldmann-Wüstefeld, T., and Vogel, E. K. (2019). Neural Evidence for the Contribution of Active Suppression During Working Memory Filtering. *Cerebral Cortex*. https://doi.org/10.1093/cercor/bhx336

Fisher, R. A. (1992). *Statistical Methods for Research Workers*. https://doi.org/10.1007/978-1-4612-4380-9_6

Foster, J. J., Sutterer, D. W., Serences, J. T., Vogel, E. K., and Awh, E. (2017). Α-Band Oscillations Enable Spatially and Temporally Resolved Tracking of Covert Spatial Attention. *Psychological Science*, *28*(7), 929–941. https://doi.org/10.1177/0956797617699167

Gulbinaite, R., İlhan, B., and Vanrullen, R. (2017). The triple-flash illusion reveals a driving role of α-band reverberations in visual perception. *Journal of Neuroscience*, *37*(30), 7219–7230. https://doi.org/10.1523/JNEUROSCI.3929-16.2017

Kasten, F. H., Wendeln, T., Stecher, H. I., and Herrmann, C. S. (2020). Hemisphere-specific, differential effects of lateralized, occipital–parietal α- versus γ-tACS on endogenous but not exogenous visual-spatial attention. *Scientific Reports*, *10*(1). https://doi.org/10.1038/s41598-020-68992-2

Pang, Z., Alamia, A., and Vanrullen, R. (2020). Turning the stimulus on and off changes the direction of α traveling waves. *ENeuro*, *7*(6), 1–11. https://doi.org/10.1523/ENEURO.0218-20.2020

Schuhmann, T., Kemmerer, S. K., Duecker, F., de Graaf, T. A., Oever, S. ten, de Weerd, P., and Sack, A. T. (2019). Left parietal tACS at α frequency induces a shift of visuospatial attention. *PLoS ONE*, *14*(11). https://doi.org/10.1371/journal.pone.0217729

Sokoliuk, R., Mayhew, S. D., Aquino, K. M., Wilson, R., Brookes, M. J., Francis, S. T., Hanslmayr, S., and Mullinger, K. J. (2019). Two spatially distinct posterior α sources fulfill different functional roles in attention. *Journal of Neuroscience*, *39*(36), 7183–7194. https://doi.org/10.1523/JNEUROSCI.1993-18.2019

Thut, G., Nietzel, A., Brandt, S. A., and Pascual-Leone, A. (2006). α-Band electroencephalographic activity over occipital cortex indexes visuospatial attention bias and predicts visual target detection. *Journal of Neuroscience*, *26*(37), 9494–9502. https://doi.org/10.1523/JNEUROSCI.0875-06.2006

Zoefel, B., Davis, M. H., Valente, G., and Riecke, L. (2019). How to test for phasic modulation of neural and behavioural responses. *NeuroImage*. https://doi.org/10.1016/j.neuroimage.2019.116175